# Last Resort from Nursing Shortage? Comparative Cost Analysis of Open vs. Robot-Assisted Partial Nephrectomies with a Focus on the Costs of Nursing Care

**DOI:** 10.3390/cancers15082291

**Published:** 2023-04-14

**Authors:** Philip Zeuschner, Carolin Böttcher, Lutz Hager, Johannes Linxweiler, Michael Stöckle, Stefan Siemer

**Affiliations:** 1Department of Urology and Pediatric Urology, Saarland University, 66123 Homburg, Saarland, Germany; 2SRH Distance Learning University, Kirchstraße 26, 88499 Riedlingen, Germany

**Keywords:** robot-assisted surgery, partial nephrectomy, nursing shortage, cost analysis, health economics

## Abstract

**Simple Summary:**

As robotic surgery is less invasive and patients recover faster compared to open surgery, the postoperative nursing effort should be lower. However, this potential cost savings mechanism for the otherwise more expensive robotic surgery has not been investigated yet. Therefore, we compared 198 robotic and 61 open partial kidney resections performed within two years at an experienced center. Indeed, the median total nursing time and daily nursing effort were significantly lower after robotic surgery, which resulted in mean savings of EUR 186.48 in nursing costs per robotic surgery. However, this cost savings mechanism alone did not amortize the overall increased costs of the robotic system.

**Abstract:**

Despite perioperative advantages, robot-assisted surgery is associated with high costs. However, the lower morbidity of robotic surgery could lead to a lower nursing workload and cost savings. In this comparative cost analysis of open retroperitoneal versus robot-assisted transperitoneal partial nephrectomies (PN), these possible cost savings, including other cost factors, were quantified. Therefore, patient, tumor characteristics, and surgical results of all PN within two years at a tertiary referral center were retrospectively analyzed. The nursing effort was quantified by the local nursing staff regulation and INPULS^®^ intensive care and performance-recording system. Out of 259 procedures, 76.4% were performed robotically. After propensity score matching, the median total nursing time (2407.8 vs. 1126.8 min, *p* < 0.001) and daily nursing effort (245.7 vs. 222.6 min, *p* = 0.025) were significantly lower after robotic surgery. This resulted in mean savings of EUR 186.48 in nursing costs per robotic case, in addition to savings of EUR 61.76 due to less frequent administrations of erythrocyte concentrates. These savings did not amortize the higher material costs for the robotic system, causing additional expenses of EUR 1311.98 per case. To conclude, the nursing effort after a robotic partial nephrectomy was significantly lower compared to open surgery; however, this previously unnoticed savings mechanism alone could not amortize the overall increased costs.

## 1. Introduction

Healthcare costs have significantly risen worldwide within the last 15 years, especially in high-income countries [1], partly caused by the increased use of robot-assisted surgical systems [2]. Since their approval in 2000, da Vinci^®^ surgical systems (Intuitive Surgical Inc., Sunnyvale, CA, USA) have spread rapidly worldwide, especially in urologic departments, so that potentially all urologic procedures can be performed with robotic assistance in specialized centers today. For instance, more than 50% of all partial nephrectomies (PN) in the United States were performed with robotic assistance in 2014 [3], and the number of robotic PNs in German hospitals has nearly doubled from 1820 in 2015 to 2989 in 2019 [4].

However, the higher acquisition and running costs compared to open surgery are often not fully reimbursed, depending on the respective healthcare system [5]. The local German Diagnosis Related Groups (DRG) system, for instance, does not cover the higher expenses for the robotic system, which is why various hospital groups did initially not provide robotic surgical systems at all. Nonetheless, a profitable use appears to be possible [6,7]. Some studies even indicate cost savings in high-volume centers compared to open surgery [5,8,9,10]. Of great importance, lower morbidity, with at least equivalent oncologic outcomes compared with the open approach, has been proven for almost all robot-assisted interventions [11,12,13,14,15,16].

Beyond the increasing economic constraints, healthcare systems worldwide have recently been subject to further burdens due to the SARS-CoV-2 pandemic which intensified many pre-existing problems. In Germany, there was an increased discourse in the general public about the working conditions of the nursing staff under the catchphrase of a “nursing emergency”. Correspondingly, the neologism “Pflexit”, the departure of nurses from their professions, was voted third place as the “Word of the Year” in 2021 in Germany [17]. The German legislation has been trying to limit this shortage of nursing staff for some time and the “nursing staff strengthening act” has substantially changed the nursing staff cost reimbursement by outsourcing the nursing staff costs from the G-DRG system in the aG-DRG catalog (a for German word for “excluded”).

However, against the background of a nursing shortage on the one hand and increasing economic constraints on the other hand, robot-assisted surgery might open up new opportunities: as robotic surgery is less invasive compared to open surgery, the postoperative nursing effort should be lower. This potential reduction of the workload for the nursing staff and this possible cost-saving mechanism has not yet been investigated and could (partially) amortize the increased material costs. To investigate this hypothesis, all retroperitoneal open vs. transperitoneal robot-assisted partial nephrectomies performed within two years at a tertiary referral center were analyzed in this retrospective, single-center study. In addition to the costs for nursing care, other cost factors were included in this comparative partial cost analysis.

## 2. Materials and Methods

All open and robot-assisted partial nephrectomies in adults performed at a tertiary referral center for robotic urologic surgery in 2020 and 2021 were retrospectively analyzed. All robotic surgeries were held using a daVinci^®^ X System (Intuitive Surgical Inc., Sunnyvale, CA, USA) via transperitoneal access, and open partial nephrectomies were performed via retroperitoneal access and a flank incision [13]. Immediately postoperatively, all patients were monitored in the urologic intermediate care unit (IMU). They were transferred to the normal ward as soon as they were cardiopulmonary stable, pain compensated, and adequately mobilized. The patients were discharged at sufficient convalescence.

For each patient, demographic factors (i.e., Charlson Comorbidity Index, CCI) and tumor specifics, such as tumor size or PADUA score depicting the tumor complexity, were obtained [15]. The histopathologic reports and intraoperative characteristics, such as warm ischemia time (WIT) were analyzed. Postoperative complications were classified according to the Clavien Dindo Classification and the number of applied erythrocyte concentrates was collected. The surgical outcome was measured using the Trifecta criteria and MIC score (margin, ischemia, and complications) defining the “success” of a partial nephrectomy (MIC: no positive surgical margins, WIT ≤ 20 min, no postoperative major complications; trifecta: no positive surgical margins, WIT ≤ 25 min, no postoperative complications) [16,17].

The postoperative nursing care effort was quantified via the German nursing staff regulation in the normal ward and via the intensive care and performance recording system (INPULS^®^) in the IMU or intensive care unit (ICU) [18]. Both systems were developed to measure the patient-specific nursing effort in different dimensions. For the INPULS system, each patient is assigned to one out of six nursing categories per 24 h with respective nursing minute values defining the actual nursing care effort (Appendix A). The INPULS system also enables the accurate recording of the occupancy time of a patient in the IMU or ICU, resulting in exact nursing minute values per patient.

For the partial cost analysis, the costs for nursing care, administration of erythrocyte concentrates, and material and consumption costs, including sterilization to perform an average robot-assisted or open PN within the analyzed period, were calculated. The German aG-DRG “L13B”, based on the local base-case value, was applied as revenue. With regard to the nursing effort, the hospital-specific nursing charge value of EUR 163.09 per day was applied based on §15 paragraph 2a of the German hospital remuneration act. For the consumables, all materials prepared for surgery by the surgical staff by the standard were assessed. The sterilization costs were calculated in sterile goods units. The analysis was carried out from a case- and occupancy-related perspective.

The statistical analysis was performed using IBM SPSS Statistics System Version 23 (International Business Corporation, Armonk, New York, NY, USA). Categorical (absolute, relative frequency) and continuous (median, range) variables were differentiated. A 1:1 nearest-neighbor propensity score matching was performed with the Charlson Comorbidity Index and PADUA score as matching variables, and the tolerance rate was set to 0.05. Group comparisons were carried out with Fisher’s exact test, chi-square, Mann–Whitney U, sign tests, McNemar, and Wilcoxon tests. All tests were two-sided and *p* < 0.05 was considered significant. This work was approved by the responsible ethics committee (Ethics Committee of the Medical Association Saarland, AZ Bu 67/19) and complies with the Declaration of Helsinki (World Medical Association, 2013).

## 3. Results

Of 259 included patients, 61 (23.6%) underwent open and 198 (76.4%) robot-assisted partial nephrectomies. They were 67.2% male and had a median age of 65 (open) vs. 63 (robotic) years (Table 1). Both groups were comparable in terms of patient characteristics. Patients undergoing open surgery were more comorbid (Charlson Comorbidity Index 10 vs. 7, *p* < 0.001), had more complex (PADUA 9 vs. 8, *p* = 0.002) and by 0.9 cm significantly larger tumors (4.2 vs. 3.3 cm, *p* = 0.004; Table 1).

After propensity-score matching, 54 patients per group were compared again without any remaining inherent group differences (Appendix A). In the matched analysis, robot-assisted partial nephrectomies lasted significantly longer (open 137.5 vs. robotic 167 min, *p* = 0.005) with a lower proportion of tumor excisions without ischemia (open 31.5% vs. 5.6%, *p* = 0.001; Table 2). The overall complication rate and severity of complications were significantly lower after robot-assisted partial nephrectomy (Table 2). The success rates of Trifecta and MIC were achieved significantly more often after robot-assisted surgery (Trifecta: open 38.9% vs. 75.9%, *p* < 0.001).

During the postoperative course, two (3.7%) patients were transferred to the ICU after open partial nephrectomy in the propensity-score-matched analysis, and none after robotic surgery (Table 3). The median length of stay of all patients undergoing open surgery in ICU was zero (range 0–9) days, with an occupancy time of 136.7 (64.5; 208.9) hours and a median nursing effort of 5339.2 (1937.4; 8861) min. On the IMU, the median length of stay after open partial nephrectomy was three (1; 17) days versus one (0; 6) day after robotic intervention (*p* < 0.001). The respective median nursing time of 1305.6 (213.1; 10931.3) min after open partial nephrectomy was more than twice as long compared to robotic surgery (*p* < 0.001). The median length of stay on the normal ward was six (2; 32) vs. four (2; 11) days and was also significantly longer in terms of the nursing effort with 803.5 (70; 4619) vs. 518 (239; 1631) min after open partial nephrectomy (*p* < 0.001). The total nursing time and mean nursing time per day were significantly longer after open partial nephrectomy (open 245.7 vs. 222.6 min, *p* = 0.025).

With average nursing staff costs for a urological nurse of EUR 0.6061 per minute, the daily nursing costs were EUR 148.92 after open vs. EUR 134.92 after robotic partial nephrectomy (Table 4). Against a revenue of EUR 130.60 nursing costs per day, this resulted in a deficit of EUR 18.32 per day after open surgery and a surplus of EUR 21.6 after robotic surgery. As a result of the shorter length of stay after robotic surgery, there was an excess revenue of EUR 186.48 per robotic partial nephrectomy in nursing costs.

At a median consumption of 0.70 erythrocyte concentrates after open and 0.28 after robotic partial nephrectomy, average costs per case of EUR 102.04 and EUR 40.28, respectively, were incurred (Appendix A). This corresponded to an excess revenue of EUR 61.76 per robotic partial nephrectomy in costs for erythrocyte concentrates. The material costs per procedure were EUR 1264.55 for robotic and EUR 124.85 for open surgery (Appendix A), and average sterilization costs were EUR 164.77 and EUR 104.02, respectively. At average maintenance costs of EUR 357.77 per robotic procedure (for a total of ca. 600 utilizations of the robotic system per year), the total costs resulted in EUR 1789.09 per robotic and EUR 228.87 per open partial nephrectomy.

From a case-related perspective, the reduced nursing effort resulted in a cost savings of EUR 186.58 per robotic partial nephrectomy, in addition to a savings of EUR 61.76 due to less frequent administrations of erythrocyte concentrates (Table 5). These cost savings did not amortize the higher material and consumption costs of EUR 1560.22; additional costs of EUR 1311.98 were incurred. In contrast, from an occupancy-related perspective, the shorter length of stay after a robotic partial nephrectomy resulted in higher revenue potential. During the median length of stay of nine days for an open partial nephrectomy, up to 1.8 times the aG-DRG revenue could be generated within the same time by the robotic procedure, since the length of stay after robotic surgery was shorter. This generated a potential occupancy-related additional revenue of EUR 5615.02 by the robotic approach (Figure 1).

## 4. Discussion

Healthcare expenditures worldwide have significantly risen over the past decade, partly due to an increased application of cost-intensive technologies [1,18]. This includes robot-assisted surgery, which is associated with high running and acquisition costs, and has superior perioperative outcomes [19]. Against the background of a significant nursing shortage in some European countries, especially in Germany, the reduced morbidity and a potentially lower nursing workload after a robotic partial nephrectomy might represent a previously unrecognized relief in nursing care and a cost-saving mechanism. To this end, we performed a partial cost analysis with a focus on the nursing care effort of all open vs. robotic partial nephrectomies within the past two years at our department. Ultimately, it was confirmed that the nursing effort was significantly lower after robotic surgery as were the case-related nursing-staff costs. Nevertheless, these savings could not compensate for the additional costs for consumables and sterilization costs associated with the robotic system.

During the analyzed period, a total of 259 partial nephrectomies were held, of which more than 2/3 were performed with robotic assistance. This can be considered a high caseload with an annual number of more than 110 robotic partial nephrectomies in 2021 [20]. The tumor complexity was rather high with nearly 40% high-risk tumors according to PADUA score in the propensity-score-matched analysis [20,21]. In the matched comparison, the operating time was longer after robotic partial nephrectomy. However, there were less frequent and less severe postoperative complications after robotic surgery, the success criteria Trifecta and MIC were achieved more frequently, and the length of stay was shorter. Hence, the results of robotic partial nephrectomy were superior to open surgery [20,22]. Of note, the length of stay after robotic partial nephrectomy underlies multiple factors and can be significantly shorter in other countries. A French working group has recently proven the safety and feasibility of robot-assisted partial nephrectomies in low-risk situations in an outpatient setting [23]. In Germany, this appears rather little attractive, since the revenue of the German diagnosis-related group “L13B” would be significantly shortened as the length of stay would fall below the lower predefined limits.

The hypothesis that the lower morbidity of robotic partial nephrectomy results in a lower nursing workload with potential cost savings could be confirmed in our analysis. The nursing effort was quantified by means of the German nursing staff regulation on the normal wards and the INPULS^®^ categories on the IMU and ICU; it was half as time and labor intensive after robotic partial nephrectomy. Against the background of the differences in length of stay, the mean nursing effort per day was still lower in the robotic group, resulting in savings of EUR 186.46 nursing staff costs per robotic surgery. Of course, it must be questioned whether the minute values adequately reflected the real nursing effort, against the background of other measurement systems for intensive care units. However, the “Therapeutic Intervention Scoring System-28 (TISS-28)” as another common classification system does not adequately reflect the true nursing effort [24,25].

In order to compare these savings with the material costs, a partial cost analysis was performed. To date, two meta-analyses comparing robotic vs. open partial nephrectomies are available. According to Wu et al., the cost difference was not significant, though the costs were slightly higher after robotic surgery [26]. Mir et al. did neither show significant differences, though laparoscopy was found to be the most cost effective [27]. In contrast, Buse et al. estimated that robotic partial nephrectomy was more expensive than open surgery, based on US data reflecting the impact of complications [28]. Shortly thereafter, the same authors concluded in another analysis that the cost of robotic surgery was approximately USD 270 lower [9]. Accordingly, Bahler et al. highlighted in their work at a longitudinal approach that robotic surgery was still USD 1464 more expensive than open surgery in 2009, though already USD 456 cheaper in 2012 [5]; the same was confirmed by Camp et al. for the United Kingdom [29]. In this regard, robotic surgery has become more cost-effective in recent years [10]. A direct comparison of these analyses with our data is difficult since healthcare systems widely differ and the cost factors analyzed here were often ignored in the above-mentioned works. Nevertheless, our estimated values for the sterilization process are similar to those of a French working group [30].

Another highly important cost factor to be analyzed in future investigations is without doubt the costs for the physician staff [31,32]. In 2021, EUR 2000.27 and therewith 28.5% of the total reimbursement for a partial nephrectomy (aG-DRG L13B) was reserved for physician staff costs. Thereof, EUR 780.53 (39%) was addressed for the surgeon during surgery, EUR 525.63 (26.3%) for the anesthetist, EUR 482.47 (24.1%) for medical work on the normal ward, and EUR 79.4 (4%) for physician care on ICU. In order to compare these cost factors between open vs. robot-assisted (vs. potentially laparoscopic) partial nephrectomy in future analyses, the true costs for the surgeon could be deviated from the skin-to-skin time in the operating room, for the anesthetist by the respective anesthesia protocols. However, it will be challenging to estimate the exact time the medical doctors spend with their patients in the normal ward or ICU; in an ideal setting, this could be measured with a stopwatch, which is challenging to implement in the daily routine. As the hourly earnings of medical staff are higher than for nursing, and the postoperative need for medical care is obviously lower after robotic surgery compared to the open intervention, this important cost factor could further reduce the financial gap between open and robotic surgery. However, this hypothesis needs to be confirmed in further work.

Nonetheless, it should be highlighted that our partial cost analysis cannot answer the question whether the robotic partial nephrectomies were *overall* cost covering. Process optimization of robotic partial nephrectomy is certainly not complete against the background of retroperitoneoscopic procedures [33] or potential physician assistants for bedside assistance. With new competitors in the field, such as the HUGO^®^ RAS system (Medtronic, Dublin, Ireland), the costs for robotic systems might continue to decrease in the future [34,35]. In addition, the German aG-DRG reimbursement system is based on cost data from so-called “costing hospitals” which increasingly perform robotic surgery themselves—which is why the reimbursement is expected to continue to increase, as it has significantly risen in recent years for robot-assisted radical prostatectomy [36].

Limitations of this study include the single center setting and a rather short time span, but at a high caseload. Furthermore, the nursing care effort was measured retrospectively. The inherent selection bias was compensated via propensity score matching but reduced the number of cases included in the analysis. Moreover, this analysis only included two specific surgical techniques for partial nephrectomy, namely the retroperitoneal open vs. the transperitoneal robot-assisted access. Partial nephrectomy can also be performed laparoscopically, either at a transperitoneal or at a retroperitoneal approach, but was not included here, since we do not perform laparoscopic partial nephrectomy in our department. Moreover, it is also possible to perform a robot-assisted partial nephrectomy in a retroperitoneoscopic fashion, which has not been established at our institute yet. These different surgical techniques should clearly be investigated in further analyses. Last but not least, only selected cost factors were analyzed—though at high precision; in particular, there are currently no comparable data on postoperative nursing care time and equivalent nursing care costs.

## 5. Conclusions

It can therefore be concluded that the primary hypothesis of this investigation, namely that the nursing workload was lower after robotic transperitoneal partial nephrectomy compared to retroperitoneal open surgery, could be confirmed. This led to a measurable relief for the nursing staff and clearly represents a previously unknown cost-saving mechanism. However, the respective cost savings, including the less-frequent administration of erythrocyte concentrates, could not amortize the increased material costs of the robotic system—although the question of a cost-covering use of the robotic system was ultimately not answered here. Nevertheless, the increased use of robotic systems in times of a “nursing emergency” in some European countries, such as Germany, could contribute to a lower migration of nursing staff in the future by reducing the daily nursing workload and labor intensity.

## Figures and Tables

**Figure 1 cancers-15-02291-f001:**
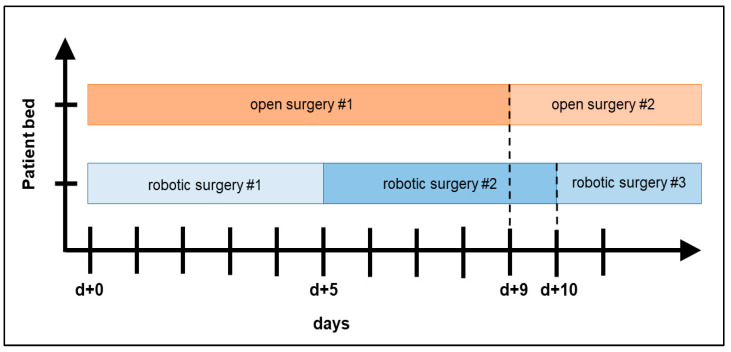
Comparison of robot-assisted vs. open partial nephrectomies from an occupancy-related perspective: in the case of ideal bed occupancy after robotic surgery, its shorter median length of stay enables additional revenue, since more patients can be operated on in the same period.

**Table 1 cancers-15-02291-t001:** Comparison of patient and tumor characteristics and the histopathological results of open vs. robot-assisted partial nephrectomies (PN) in the overall analysis.

	Open PN (n = 61)	Robotic PN (n = 198)	*p*-Value
**patient characteristics**			
age	65 (23; 90)	63 (26; 85)	0.151
gender male	41 (67.2%)	133 (67.2%)	1.000
size [cm]	171.5 (152; 195)	174.5 (150; 195)	0.085
weight [kg]	85 (51; 145)	84 (45; 142)	0.939
body-mass index [kg/m²]	27.2 (20.9; 49.6)	28.0 (17.8; 53.0)	0.413
Charlson comorbidity score (CCI)	10 (2; 19)	7 (1; 20)	<0.001
**tumor specifics**			
PADUA score	9 (6; 13)	8 (6; 12)	0.002
PADUA low risk (6, 7)	11 (18.0%)	55 (27.8%)	0.135
PADUA mid risk (8, 9)	21 (34.4%)	101 (51.0%)	0.028
PADUA high risk (≥10)	29 (47.5%)	42 (21.2%)	<0.001
tumor size [cm]	4.2 (0.9; 12.0)	3.3 (0.5; 9.0)	0.004
**malignancy**	44 (72.1%)	148 (74.7%)	0.559
pT			0.240
pT1a	25 (41.0%)	93 (47.0%)	0.463
pT1b	10 (16.4%)	42 (21.2%)	0.468
pT2a	2 (3.3%)	2 (1.0%)	0.468
pT3	7 (11.5%)	10 (5.1%)	0.134
pN1	1 (1.6%)	1 (0.5%)	0.774
pR1	6 (9.8%)	16 (8.1%)	0.747
**histological subtype**			0.093
clear cell	35 (57.4%)	97 (49.0%)	0.305
chromophobe	3 (4.9%)	14 (7.1%)	0.769
papillary	5 (8.2%)	35 (17.7%)	0.103
other malignant	1 (1.6%)	2 (1.0%)	0.555
angiomyolipoma	2 (3.3%)	7 (3.5%)	1.000
oncocytoma	9 (14.8%)	39 (19.7%)	0.454
cyst	2 (3.3%)	1 (0.5%)	0.139

**Table 2 cancers-15-02291-t002:** Comparison of the perioperative outcomes of open vs. robot-assisted partial nephrectomies (PN) after propensity score matching.

	Open PN (n = 54)	Robotic PN (n = 54)	*p*-Value
**intraoperative results**			
operating time [min]	137.5 (63.5; 286)	167 (77; 342)	0.005
blood loss [mL]	300 (50; 2200)	250 (40; 1500)	0.777
off-clamp excisions	17 (31.5%)	3 (5.6%)	0.001
WIT [min]	15 (5; 30)	13.5 (7; 46)	0.934
conversions			
to open surgery	-	-	
to nephrectomy	3 (5.6%)	0	0.250
**postoperative results**			
complications	28 (51.9%)	9 (16.7%)	<0.001
Clavien Dindo grade 1	2 (3.7%)	1 (1.9%)	1.000
grade 2	7 (13.0%)	3 (5.6%)	0.344
grade 3a	5 (9.3%)	3 (5.6%)	0.727
grade 3b	11 (20.4%)	1 (1.9%)	0.006
grade 4a	3 (5.6%)	1 (1.9%)	0.625
grade 5	-	-	1.000
erythrocyte concentrate yes	10 (18.5%)	4 (7.4%)	0.146
intraoperative number	0 (0; 1)	0 (0; 2)	0.414
postoperative number	0 (0; 11)	0 (0; 7)	0.082
total number	0 (0; 11)	0 (0; 7)	0.160
Trifecta fulfilled	21 (38.9%)	41 (75.9%)	<0.001
MIC fulfilled	26 (48.1%)	40 (74.1%)	0.008

**Table 3 cancers-15-02291-t003:** Comparison of the nursing care effort of open vs. robot-assisted partial nephrectomies (PN) in the propensity-score-matched analysis.

	Open PN (n = 54)	Robotic PN (n = 54)	*p*-Value
**length of stay [d]**			
ICU	0 (0; 9)	-	<0.001
IMU	3 (1; 17)	1 (0; 6)	<0.001
normal ward	6 (2; 32)	4 (2; 11)	<0.001
total	9 (5; 36)	5 (3; 15)	<0.001
**occupancy time [h]**			
ICU	136.7 (64.5; 208.93)	-	<0.001
IMU	47.5 (7.8; 397.5)	22.1 (0; 140.8)	<0.001
**nursing time [min]**			
ICU	5399.2 (1937.4; 8861)	-	<0.001
IMU	1305.6 (213.1; 10,931.3)	605 (0; 3.901.5)	<0.001
normal ward	803.5 (70; 4.619)	518 (239; 1631)	<0.001
total	2407.8 (995.1; 12,599.3)	1126.8 (656; 4626.5)	<0.001
**nursing time per day [min]**	245.7 (160.3; 659.2)	222.6 (131.2; 513.7)	0.025

**Table 4 cancers-15-02291-t004:** Comparison of the revenue for nursing care of open vs. robot-assisted partial nephrectomies (PN) in the propensity-score-matched analysis.

	Open PN (n = 54)	Robotic PN (n = 54)
nursing care time/day	245.7 min	222.6 min
nursing costs/day ^1^	EUR 148.92	EUR 134.92
revenue/day ^2^	EUR −18.32	EUR +4.32
**total profit ^3^**	EUR −164.88	EUR +21.6

^1^ based on EUR 0.6061 / nursing minute; ^2^ based on EUR 130.60 nursing revenue / day; ^3^ based on length of stay open PN 9, robotic PN 5 days.

**Table 5 cancers-15-02291-t005:** Overview of the case-related savings potential of robotic partial nephrectomy.

	Potential Cost Savings
nursing-care costs	EUR +186.48
erythrocyte concentrates	EUR +61.76
material and consumption costs	EUR −1560.22
**sum**	EUR −1311.98

## Data Availability

The data that support the findings of this study are available from the corresponding author upon reasonable request.

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
