# Peer review of "Last Resort from Nursing Shortage? Comparative Cost Analysis of Open vs. Robot-Assisted Partial Nephrectomies with a Focus on the Costs of Nursing Care"

_cancers, 2023, doi:10.3390/cancers15082291_

Round 1

Reviewer 1 Report

The authors compare robotic vs open surgery in a single center setting for partial nephrectomies focusing on potential savings in nursing costs.

The paper is well designed with one exception that needs to be explained and discussed. The results are straightforward presented and thoroughly discussed and balanced, particularly looking into the overall costs which are still not in favor for robotic surgery, though the gap is closing.

This unveils a weakness in the paper that needs to be addressed in introduction, discussion and conclusion since the authors choose not to add retroperitoneoscopic/laparoscopic procedures as control group. Nevertheless I would still favor publication since open access for partial nephrectomies is still a dominant approach in Germany.

Major point:

Why is robotic surgery not compared to retroperitoneoscopic/laparoscopic surgery?

Reviewer 2 Report

The authors investigated the cost saving mechanisms focusing on the costs of nursing care and clarified that RAPN could reduce the nursing care cost, however, it did not amortize the overall costs of RAPN.

Questions

1.       As authors mention physician staff costs, I think it should be discussed more because physicians are involved in not only IMU/ICU/wards but also procedures during surgery. Usually their remuneration is higher than other medical staffs. That should increase the total costs of open PN.

2.       Please show or mention the oncological outcomes, such as recurrence rate and survivals.

3.       Please explain why authors compared transperitoneal access RAPN and retroperitoneal open PN, not select retro- RAPN, transperitoneal PN, and/or laparoscopic PNs.
